# Microbiome and pediatric leukemia, diabetes, and allergies: Systematic review and meta-analysis

Rachel Gallant[1,2]*, Samiha Reza[1], Joseph L. Wiemels[1], Mel Greaves[3]

**1** Center for Genetic Epidemiology, Department of Population and Public Health Sciences, University of Southern California Keck School of Medicine, Los Angeles, California, United States of America, **2** Pediatric Hematology-Oncology, University of Oklahoma Health Sciences Center, Oklahoma City, Oklahoma, United States of America, **3** Centre for Evolution and Cancer, The Institute of Cancer Research, London, United Kingdom

* Rachel-gallant@ouhsc.edu

## Abstract

### Background

Despite the different pathologies and genetic susceptibilities of childhood ALL, T1DM and allergies, these conditions share epidemiological risk factors related to timing of infectious exposures and acquisition of the gut microbiome in infancy. We have assessed whether *lower* microbiome diversity (Shannon Index) and shared genus/species profiles are associated with pediatric ALL, allergies, and T1DM.

### Methods and findings

Literature search was performed using PubMed, Embase, Cochrane, and Web of Science databases. Case-control, meta-analyses, and cohort studies were considered for inclusion. Inclusion criteria: (i) subjects age 1–18 years at diagnosis, (ii) reports effect of microbiome measured prior to/at time of diagnosis/first intervention (iii) outcome of ALL, allergies, asthma, or T1DM, (iv) English text. Exclusion criteria: (i) age < 1 or >18 years at diagnosis, (ii) Down Syndrome-associated ALL, (iii) non-English text, (iv) reviews, pre-print, or abstracts, (v) heavily biased studies. Abstract and full text screening were performed by two independent reviewers. Data extraction was performed by one reviewer following PRISMA guidelines. Data were pooled using a random-effects model. Eighty-eight studies were included in the analysis, with seventy-seven in the qualitative analysis and 54 in the meta-analysis. Cases were found to have lower alpha-diversity than controls in ALL (SMD:-0.78, 95%CI:-1.21, -0.34), T1DM (SMD:-1.26, 95%CI:-3.49, 0.96), eczema (SMD:-0.34, 95%CI:-0.56, -0.12), atopy (SMD:-0.06, 95%CI:-0.34, 0.22), asthma (SMD:-0.37, 95%CI:-1.16, 0.42), and food allergy (SMD:-0.11, 95%CI:-0.63, 0.41).

**Data availability statement:** All relevant data are within the manuscript and its Supporting Information files.

**Funding:** This work was supported by Cancer Research UK (CRM 171X). The funders had no role in study design, data collection and analysis, decision to publish, or preparation of the manuscript.

**Competing interests:** The authors have declared that no competing interests exist.

## Conclusions

These results highlight similarities in the microbiome diversity and composition of children with ALL, T1DM, and allergies. This is compatible with a common risk factor related to immune priming in infancy and highlights the gut microbiome as a potentially modifiable risk factor and preventative strategy for these childhood diseases.

## Introduction

Microbial exposures early in life are important for developing a competent immune system, and a lack of such exposure may result in dysregulated immune responses to subsequent antigenic stimuli [1–3]. The gut microbiome is measurable by way of fecal samples or rectal swabs and differences in its composition have been noted when comparing delivery modes, breastfeeding, and other surrogates for microbial exposure. These differences can persist throughout infancy and early childhood suggesting that early microbial exposures have a lasting impact on immune development in children and thereby the risk of childhood diseases such as ALL, T1DM, and allergies [4,5]. Notably the gut microbiome can be modified by administration of probiotics, fecal transplantation, and diet in addition to the environmental exposures previously mentioned making it an important potential target for disease prevention measures [6–9].

Acute lymphoblastic leukemia (ALL) is the most common childhood malignancy [10]. Though most children with ALL experience a long-term remission with modern therapy, patients with high risk molecular features and patients with relapsed disease experience much lower survival rates [10]. Those who do achieve a sustained remission are subject to the acute and chronic toxicities of chemotherapy as well as the psychological and financial burden of undergoing prolonged therapy [10]. This makes the question of ALL etiology highly relevant, because a better understanding of how ALL develops could provide approaches to *prevent* cases of ALL. Indeed, the inexorable rise in incidence of ALL over the past 70 years strongly implicates modifiable environmental risk factors in causation [5,11–13].

ALL etiology is likely multifactorial, and the risk factors and drivers of disease may differ among subtypes of pediatric ALL. Nonetheless, it is known that acquired genetic aberrations associated with ALL (such as *ETV6::RUNX1*) are present at birth in a much larger proportion of children than overt leukemia [14,15]. This indicates that these pre-leukemic clones do not progress to overt leukemia without subsequent, post-natal acquisition of secondary genetic changes. Common infectious exposures, including respiratory viruses, are believed to trigger the essential secondary genetic changes [4] primarily via B-cell recombinase (RAG) mediated gene deletions [16]. But the latter, operating via a dysregulated immune response, is suggested to be contingent upon a deficit of microbial exposures and immune priming in infancy [4]. The epidemiological risk factors indicative of early life microbiome exposure can be considered as surrogates for acquisition of the gut microbiome. For instance, C-section delivery and being first-born (both indicative of lower microbial exposure) are

risk factors for ALL, whereas breastfeeding and daycare attendance (higher microbial exposure) are protective [17–23]. These data support the hypothesis that increased microbial exposure in infancy leading to appropriate immune development is associated with decreased risk of ALL development whereas those with less microbial exposure may have more potential for future immune dysregulation, or chronic inflammation, thereby increasing their risk of ALL [4,14,17].

Epidemiologic evidence suggests that early life microbial exposures also impact the risk of type 1 diabetes mellitus (T1DM) and allergic disorders of childhood such as asthma, eczema, and allergies [3,24–27]. Like ALL, T1DM and allergic conditions are associated with unique genetic susceptibilities that alone do not result in overt disease without additional environmental triggers.

Despite the different pathologies and background genetic susceptibilities these three childhood illnesses may share a common immune priming deficiency contingent upon a lack of a diverse gut microbiome. If correct, this should be reflected in a consistent microbiome composition. Therefore, in this systematic review and meta-analysis we aim to identify similarities and differences in microbiome composition between ALL, T1DM, and allergic disorders of childhood.

## Methods

We performed a systematic review and meta-analysis examining the effect of microbiome diversity on the development of pediatric ALL, allergies, and T1DM following Preferred Reporting Items for Systematic Reviews and Meta-Analyses (PRISMA) guidelines. The review is not registered, but the complete protocol is included in S1 Appendix in S1 File. Initial literature search was performed using PubMed, Embase, Cochrane, and Web of Science databases in 2022 and an update search performed in 2024. Search terms used at both time points were: (i) (pediatric AND microbiome AND leukemia) (ii) (pediatric AND microbiome AND allergies) (iii) (pediatric AND microbiome AND type 1 diabetes). A complete list of search terms is included in S2 Appendix in S1 File. References were uploaded into EndNote software and deduplication was performed systematically [28,29]. Covidence software was used for screening, review, and data extraction [30].

### Inclusion and exclusion criteria

Case-control studies, meta-analyses, and cohort studies were considered for inclusion. Inclusion criteria were: (i) subjects age 0–18 years at diagnosis, (ii) reports the effect of microbiome as measured prior to or at the time of diagnosis/ first intervention for disease (iii) includes outcome of ALL, allergies, asthma, or type 1 diabetes, (iv) published in English. Exclusions criteria were: (i) age > 18 years at diagnosis, (ii) outcome of Down Syndrome associated ALL or infant ALL (iii) non-English text, (iv) reviews, pre-print, or abstracts, (v) heavily biased studies or those with obvious confounders.

### Selection process

Two independent reviewers (RG and SR) performed title and abstract screening of all titles returned using the search strategy outlined above, then performed full text screening independently. Disagreements were discussed until a consensus was reached. Studies not meeting inclusion criteria were excluded during this process.

### Data extraction process

Data collection forms were created using Covidence software [30]. and data was collected from each study by one reviewer (RG). Basic information including title, year of publication, country, study design, and number of participants were collected. Measures of the microbiome were collected as microbial diversity and/or abundance of specific bacterial genera. For the former, the metric used was recorded (e.g., Shannon, Chao1, Simpson, or other measure of diversity), and for the latter the bacterial genus of interest and relative abundance were recorded. Summary statistics for Shannon Diversity Index comparing cases and controls were extracted including mean and standard deviation (SD), median and 95% confidence intervals (95% CI), or regression coefficients. We also collected demographic variables including a description

of the population, race, and ethnicity when applicable. Additional variables were collected as available, including mode of delivery (vaginal vs c-section), breastfeeding (duration of breastfeeding), antibiotic exposure in infancy, maternal antibiotic exposure during pregnancy, exposure to farm animals or pets in infancy, birth order, and daycare attendance. A complete list of variables can be found in S3 Appendix in S1 File.

## Outcomes

The primary outcome is the identification of the effect of microbiome diversity on the risk of pediatric ALL, allergies, and T1DM; specifically, with the hypothesis that *lower* microbiome diversity is associated with ALL, allergic conditions, and T1DM when compared to healthy controls. We also explored associations between bacterial genera commonly reported across studies and the diseases of interest. Additionally, we considered the effects of ethnicity/country of origin, breast-feeding exposure, delivery mode, antibiotic exposure, birth order, and farm animal or pet exposure as applicable on the relationship of microbiome and the diseases of interest.

## Statistical analysis

Studies reporting Shannon Index Diversity Score (alpha-diversity) for cases and controls with appropriate summary statistics were included in the meta-analysis. Standard mean differences (SMD) were calculated for all studies. For studies reporting median and 95% CI, mean and SD were estimated to calculate SMD [31]. Regression coefficients were converted to SMD using the "esc" package in R Studio (v4.17-0). For articles that did not explicitly state the necessary summary statistics for inclusion in meta-analysis, an email was sent to the corresponding author(s) requesting this infor-mation. Authors responded either by providing the summary statistics or appropriate data to calculate summary statistics or stating that summary statistics for the analysis were no longer in their possession/available. For those authors who did not respond, a second email was sent requesting summary statistics. If summary statistics were not available in the manuscripts or directly from authors, they were estimated from graphical depictions of the results as available. If summary statistics were not available and could not be estimated graphically, then they were not included in the meta-analysis.

Individual meta-analyses were performed using the meta package in R Studio (v4.17-0) [32] for each disease type using mean Shannon Index and standard deviation. Weighted estimates were calculated based on the size of the study. Microbiome diversity (Shannon Index) was included as the predictor variable with disease (ALL, T1DM, atopy, eczema, asthma, or food allergy) as the outcome variable. Subgroup analyses were also performed in an attempt to eliminate some of the heterogeneity of study designs. Given variability between studies, a random effects model was used. Funnel plots were analyzed to assess for publication bias for studies included in each meta-analysis. The ROBIS tool was used to assess risk of bias. GRADE guidelines were followed to assess quality of the studies included in the meta-analyses (S2 Table in S1 File) [33].

Meta-analyses were not performed to evaluate the association between bacterial genera and disease outcomes due to the variability in bacterial abundance reporting between studies; instead, we performed an exploratory qualitative analysis. Relative abundance was extracted from studies describing microbiome composition and was recorded as higher or lower in cases compared to controls. The percentage of studies reporting increased or decreased abundance was calculated for each bacterial genus. This data was tabulated and clustering analysis was performed using the ComplexHeatmap package in R [34,35].

## Results

A literature search identified 6,527 articles, and after de-duplication 4,262 unique articles were available for screening (Fig 1). Two independent reviewers performed the title and abstract screening, leaving 326 articles for full-text screening. Ultimately 88 articles met inclusion criteria for qualitative and/or quantitative analysis (Fig 1).

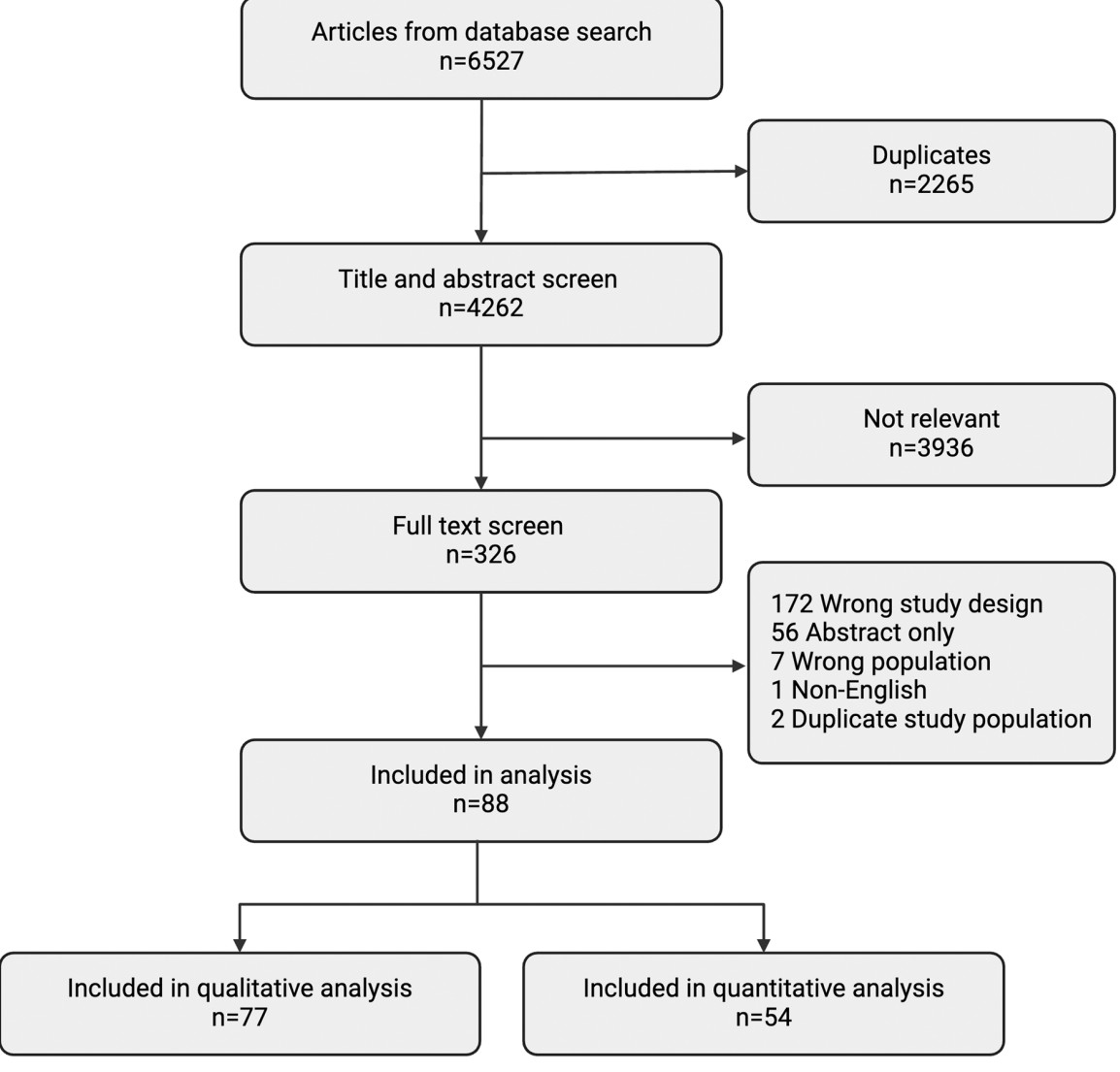

**Fig 1. Flow chart of study selection.** Created with Biorender.com.

### Alpha-diversity: Meta-analysis

Seventy-one studies reported alpha-diversity, and 54 were included in the meta-analyses (Table 1). Funnel plot analysis showed no evidence of publication bias in the studies included in the meta-analyses (S1 Fig in S1 File).

**ALL.** Six studies measuring alpha-diversity of the gut microbiome in ALL cases at diagnosis and controls were included in the meta-analysis [36–42]. ALL cases were found to have significantly lower alpha-diversity compared to controls with a standard mean difference (SMD) of -0.78 (95%CI:-1.21, -0.34, Fig 2). As recent antibiotic exposure may be expected to alter the gut microbiome composition, we performed an additional analysis excluding cases with antibiotic exposure which revealed a similar result as the overall analysis (SMD:-0.83, 95% CI:-1.27, -0.39, S2 Fig in S1 File).

**Type 1 diabetes mellitus.** Eleven studies were included in the meta-analysis of T1DM cases and controls [24,43–52]. There was no statistically significant difference in the gut microbiome alpha-diversity of T1DM cases compared to controls (SMD:-1.26, 95%CI:-3.49, 0.96). Though the difference was not significant, cases tended to have

**Table 1. Summary characteristics of identified studies reporting alpha diversity in cases vs controls.**

| Disease | Study | Study Type | Country | N | Age Mean/ Median (Range) | % Female | Sequencing | Alpha Diversity Measure | Alpha Diversity |
|---|---|---|---|---|---|---|---|---|---|
| ALL | Bai (2017)* | Case-control | China | 63 | 5 yr (1–12.7yr) | 41% | 16s rRNA (V3-V4) | Shannon Simpson† | ↓⁺ ↓⁺ |
| ALL | Chua (2020)* | Case-control | Malaysia | 14 | NA (2–6 yr) | 14% | 16s rRNA (V4) | Shannon Chao1 | ↓ ↓ |
| ALL | DePietri (2020)* | Case-control | Denmark | 70 | 3.7 yr (2.7–6.8yr) | 13% | 16s rRNA (V3-V4) | Shannon | ↓⁺ |
| ALL | Gao (2020)* | Case-control | China | 36 | 6yr | 47% | 16s rRNA (V3-V4) | Shannon Chao1 ACE Simpson† | ↓⁺ ↓⁺ ↓⁺ ↓⁺ |
| ALL | Liu (2020)* | Case-control | China | 81 | 7yr (3–16yr) | 44% | 16s rRNA (V1-V9) | Shannon Chao1 ACE Simpson | ↓ ↓ ↓ ↓ |
| ALL | Rajagopala (2020)* | Case-control | United States | 52 | 5.5yr (1–19yr) | 48% | 16s rRNA (V4) | Shannon Chao1 | ↓⁺ ↓⁺ |
| T1DM | Biassoni (2020) | Case-control | Italy | 56 | 10yr | 35% | 16s rRNA (V2-V4, V6-V9) | Shannon Chao1 ACE | ↓⁺ ↓⁺ ↓⁺ |
| T1DM | Cinek (2017)* | Case-control | Finland | 36 | 9mo | 61% | 16s rRNA (V4) | Shannon Chao1 ACE Simpson (r) | ↓ ↓ ↓ ↓ |
| T1DM | Cinek (2018) | Case-control | Nigeria, Sudan, Azerbaijan, Jordan | 176 | 11yr | 50% | 16s rRNA (V4) | Shannon Chao1 ACE Simpson | = = = = |
| T1DM | Giongo (2011) | Case-control | Finland | 8 | 2yr | NA | 16s rRNA | Shannon | ↓⁺ |
| T1DM | Harbison (2019) | Cohort | Australia | 88 | 10.9yr | 47% | 16s rRNA (V4) | Simpson (r) | = |
| T1DM | Kostic (2015) | Case-control | Estonia & Finland | 33 | 2.7yr | NA | 16s rRNA (V4) | Chao1 | ↓⁺ |
| T1DM | Leiva-Gea (2018)* | Case-control | Spain | 28 | 12yr | 50% | 16s rRNA | Shannon Chao1 | ↓⁺ ↓ |
| T1DM | Singh (2021)* | Case-control | Qatar | 30 | 14yr | NA | 16s rRNA (V3-V4) | Shannon Chao1 | ↓ ↓ |
| T1DM | Traversi (2020)* | Case-control | Italy | 96 | 8yr | 29% | 16s rRNA (V3-V4) | Shannon Simpson† | ↓ ↓ |
| T1DM | Vatanen (2018)* | Case-control | US, Finland, Germany, Sweden | 158 | 4mo | NA | Metagenomic sequencing | Shannon | ↑⁺ |
| T1DM | Chierico (2022)* | Case-control | Italy | 109 | NA (5–15yr) | 50% | 16s rRNA | Shannon Chao1 Simpson | ↑⁺ ↑⁺ ↑⁺ |
| T1DM | Xu (2022)* | Case-control | China | 118 | 8yr | 42% | 16s rRNA (V3-V4) | Shannon Chao1 | ↓⁺ ↓⁺ |
| T1DM | Belteky (2023)* | Cohort | Sweden | 284 | 1yr | 50% | 16s rRNA (V3-V4) | Shannon | ↓ |
| T1DM | Yuan (2022)* | Case-control | China | 141 | 7.7yr | 41% | 16s rRNA (V3-V4) | Shannon Chao1 | ↓⁺ ↓⁺ |
| T1DM | Elsherbiny (2022)* | Case-control | Egypt | 48 | 7.4yr | 44% | 16s rRNA (V3-V4) | Shannon Chao1 | ↓⁺ ↓⁺ |

*(Continued)*

| Disease | Study | Study Type | Country | N | Age Mean/ Median (Range) | % Female | Sequencing | Alpha Diversity Measure | Alpha Diversity |
|---|---|---|---|---|---|---|---|---|---|
| T1DM | Mokhtari (2023)* | Case-control | United States | 65 | 14yr | 46% | Metagenomic sequencing | Shannon Chao1 Simpson | ↓ ↓ ↓ |
| Eczema | Abrahamsson (2012)* | Case-control | Sweden | 40 | 12mo | 45% | 16s rRNA (V3-V4) | Shannon | ↑ |
| Eczema | Chan (2020) | Case-control | China | 48 | 4mo | 52% | 16s rRNA (V3-V4) | #Observed species | ↑ |
| Eczema | Forno (2008)* | Case-control | United States | 37 | 4mo | 48% | DGGE (V2-V3) | Shannon | ↓+ |
| Eczema | Galazzo (2020) | Case-control | Germany | 440 | 7.8mo | 49% | 16s rRNA (V3) | Shannon | ↓+ |
| Eczema | Kang (2021)* | Case-control | Korea | 160 | 6mo | 42% | 16s rRNA (V1-V3) | Shannon Chao1 | ↓ ↑ |
| Eczema | Lee (2018)* | Case-control | Korea | 129 | 6mo | 47% | 16s rRNA (V1-V3) | Shannon | ↓ |
| Eczema | Los-Rycharska (2021) | Cross-sectional | Poland | 33 | 4mo | 42% | 16s rRNA (V3-V4) | Shannon | ↓ |
| Eczema | Nylund (2015) | Case-control | Finland | 39 | 6.6mo | NA | HITChip analysis | Simpson (r) | ↓+ |
| Eczema | Park (2020)* | Case-control | Korea | 132 | 6mo | 45% | 16s rRNA (V1-V3) | Shannon | ↓ |
| Eczema | Reddel (2019)* | Case-control | Italy | 37 | 2yr | 38% | 16s rRNA (V1-V3) | Shannon Chao1 | ↓ ↓ |
| Eczema | Song (2016)* | Case-control | Korea | 132 | NA | 49% | 16s rRNA (V1-V2) | Shannon | ↓ |
| Eczema | Tang (2016)* | Case-control | China | 25 | 1mo | NA | NGS | Shannon | ↓ |
| Eczema | Wang (2008)* | Case-control | Sweden, Great Britain, Italy | 35 | 1wk | NA | T-RFLP (V3-V4, V9) | Shannon Simpson (r) | ↓+ ↓+ |
| Eczema | West (2015)* | Case-control | Australia | 20 | 1yr | 65% | 16s rRNA (V3-V4) | Shannon | ↓ |
| Eczema | Patumcharoen-pol (2023)* | Cohort | Thailand | 62 | NA (9–12mo) | 39% | 16s rRNA (V3-V4) | Shannon Chao1 Simpson | = ↓ = |
| Eczema | Sasaki (2022)* | Case-control | Switzerland | 66 | 3mo | 50% | 16s rRNA (V4) | Shannon | ↓ |
| Eczema | Loo (2022)* | Cohort | Singapore | 322 | 2yr | 45% | 16s rRNA (V4) | Shannon | ↓+ |
| Eczema | Fan (2022)* | Cohort | China | 36 | 1yr | 69% | 16s rRNA (V3-V4) | Shannon Simpson | ↑ ↓ |
| Eczema | Sung (2022)* | Cohort | South Korea | 15 | 1yr | 40% | 16s rRNA (V3-V4) | Shannon | ↓ |
| Eczema | Hoskinson (2023)* | Cohort | Canada | 805 | 1yr | 47% | Metagenomic sequencing | Shannon | ↓ |
| Atopy | Abrahamsson (2014) | Case-control | Sweden | 41 | 1yr | NA | 16s rRNA (V3-V4) | Shannon | ↑ |
| Atopy | Arrieta (2015)* | Case-control | Canada | 96 | 3mo | NA | 16s rRNA (V3) | Shannon | ↓ |
| Atopy | Arrieta (2018)* | Case-control | Ecuador | 97 | 3mo | 54% | 16s rRNA (V4) | Chao1 | ↑ |

*(Continued)*

| Disease | Study | Study Type | Country | N | Age Mean/ Median (Range) | % Female | Sequencing | Alpha Diversity Measure | Alpha Diversity |
|---|---|---|---|---|---|---|---|---|---|
| Atopy | Chiu (2020)* | Case-control | Taiwan | 42 | 4yr | 43% | 16s rRNA (V3-V4) | Shannon Chao1 | ↑ = |
| Atopy | Shen (2019)* | Case-control | China | 39 | 1yr | 56% | 16s rRNA (V3-V4) | Shannon Chao1 ACE Simpson | ↑ ↑ ↑ ↑ |
| Atopy | Simonyte-Sjodin (2019)* | Case-control | Sweden | 72 | 8yr | 45% | 16s rRNA | Shannon | ↓ |
| Atopy | Hoskinson (2023)* | Cohort | Canada | 664 | 1yr | 47% | Metagenomic sequencing | Shannon | = |
| Atopy | Wan (2023)* | Case-control | China | 37 | 9yr | 43% | Metagenomic sequencing | Shannon Simpson | ↑ ↑ |
| Atopy | Chiu (2023)* | Case-control | Taiwan | 53 | 5.7yr | 47% | Metagenomic sequencing | Shannon | ↓ |
| Asthma | Abrahamsson (2014)* | Case-control | Sweden | 36 | 1yr | NA | 16s rRNA (V3-V4) | Shannon | ↑ |
| Asthma | Chiu (2020)* | Case-control | Taiwan | 40 | 4yr | 45% | 16s rRNA (V3-V4) | Shannon Chao1 | ↓ ↓ |
| Asthma | Patrick (2020) | Case-control | Canada | 570 | 1yr | 60% | 16s rRNA (V4) | Chao1 | ↓+ |
| Asthma | Stiemsma (2016)* | Case-control | Canada | 76 | 1yr | 46% | 16s rRNA (V3) | Shannon | = |
| Asthma | Mo (2022)* | Case-control | China | 34 | 6yr | 36% | 16s rRNA (V3-V4) | Shannon Chao1 | ↓+ ↓+ |
| Asthma | Zheng (2022) | Case-control | China | 57 | 9yr | 57% | 16s rRNA (V4) | Chao1 Simpson | ↑ ↓+ |
| Asthma | Hoskinson (2023)* | Cohort | Canada | 650 | 1yr | 47% | Metagenomic sequencing | Shannon | ↓ |
| Asthma | Wan (2023)* | Case-control | China | 42 | 8.4yr | 38% | Metagenomic sequencing | Shannon Simpson | ↑+ ↑+ |
| Asthma | Chiu (2023)* | Case-control | Taiwan | 59 | 5.6yr | 41% | Metagenomic sequencing | Shannon | ↓+ |
| Atopy/ Eczema/ Asthma | Low (2017) | Case-control | China | 39 | 6mo | 26% | 16s rRNA (V3-V5) | Chao1 ACE | ↓ ↓ |
| FA | Azad (2015)* | Case-control | Canada | 166 | 1yr | 49% | 16s rRNA (V4) | Shannon Chao1 | ↑ ↑ |
| FA | Chen (2016) | Case-control | Taiwan | 45 | 13mo | 47% | 16s rRNA (V3-V5) | Shannon Chao1 | ↓+ ↓+ |
| FA | Fazlollahi (2018)* | Case-control | United States | 141 | 9.7mo | 33% | 16s rRNA (V4) | Shannon Chao1 | ↑+ ↑+ |
| FA | Inoue (2017)* | Case-control | Japan | 8 | NA 18mo-6yr | 50% | 16s rRNA (V3-V4) | Shannon Chao1 | = ↑ |
| FA | Kouroush (2018)* | Case-control | United States | 43 | NA 0-18yr | 51% | 16s rRNA (V4) | Shannon | ↑ |
| FA | Los-Rycharska (2021) | Cross-sectional | Poland | 44 | 4mo | 41% | 16s rRNA (V3-V4) | Shannon | = |
| FA | Savage (2018)* | Case-control | United States | 216 | 5mo | 49% | 16s rRNA (V3-V5) | Shannon Chao1 | ↑ ↑ |

*(Continued)*

**Table 1.** (Continued)

| Disease | Study | Study Type | Country | N | Age Mean/ Median (Range) | % Female | Sequencing | Alpha Diversity Measure | Alpha Diversity |
|---------|-------|-----------|---------|---|--------------------------|----------|------------|-------------------------|-----------------|
| FA | Yan (2023)* | Case-control | China | 20 | 1.2yr (cases) | 30% (cases) | 16s rDNA (V3-V4) | Shannon Chao1 ACE Simpson | ↓ ↓ ↓ ↓ |
| FA | Hara (2024)* | Case-control | Japan | 48 | *NA* (17–19mo) | *NA* | 16s rRNA (V3-V4) | Shannon | ↑ |
| FA | Kanchongkitti-phon (2024)* | Cross-sectional | Thailand | 60 | 3.7yr | 37% | 16s rDNA (V3-V4) | Shannon | ↓ |
| FA | Mera-Berriatua (2022) | Case-control | Spain | 50 | 5mo | 58% | 16s rRNA (V3-V4) | Shannon | ↓⁺ |
| FA | De Paepe (2024) | Case-control | Belgium | 81 | 2yr | *NA* | 16s rRNA | Chao1 | ↓⁺ |
| FA | Wang (2022) | Cohort | China | 68 | 1mo | 46% | 16s rRNA (V3-V4) | Chao1 | ↓⁺ |
| FA | Joseph (2022)* | Cohort | United States | 447 | 1-6mo | 47% | 16s rRNA (V4) | Shannon | ↓⁺ |
| FA | Chen (2024)* | Case-control | Taiwan | 81 | 2yr | 53% | Pyrose-quencing | Shannon Chao1 | ↓⁺ ↓⁺ |
| FA | Yang (2022)* | Case-control | China | 225 | 8.5yr | 46% | 16s rRNA (V3-V4) | Shannon | ↓⁺ |
| FA | Hoskinson (2023)* | Cohort | Canada | 623 | 1yr | 47% | Metagenomic sequencing | Shannon | ↓ |

Table includes studies identified from systematic review that report alpha diversity in cases compared to controls for disease types of acute lymphoblastic leukemia (ALL), T1DM (type 1 diabetes mellitus), eczema, atopy, asthma, and food allergy. ⁺ Indicates that the results of the individual study reached statistical significance with a p-value of <0.05. Simpson† indicates Simpson's Diversity Index (D). Simpson (r) indicates Simpson's Reciprocal Index (1/D).

ACE: abundance-based coverage estimator, ALL: acute lymphoblastic leukemia, DGGE: denatured gradient gel electrophoresis, FA: food allergy, T1DM: type 1 diabetes mellitus.

*Studies with available summary statistics allowing for inclusion in meta-analysis.

lower alpha diversity than controls. There were several limitations to the T1DM analysis including differences in study design, timing of microbiome assessment, and microbiome diversity measures reported that likely contribute to the lack of statistically significant results. Two of the studies included in the meta-analysis were prospective studies in which cases and controls were recruited based on having HLA haplotypes conferring a risk of T1DM development [24,46]. An analysis excluding these studies shows a stronger difference in alpha-diversity between cases and controls, though not statistically significant (S3 Fig in S1 File). Furthermore, in four of the studies some patients had already been started on insulin therapy when samples for microbiome assessment were obtained rather than at diagnosis prior to treatment [43,45]. Exclusion of those studies from the meta-analysis shows a trend in the same direction as the overall analysis (S4 Fig in S1 File). We also performed an analysis including only those studies that measured microbiome at the time of T1DM diagnosis which is more similar to the design of the ALL studies meta-analysis and shows a slight trend in the same direction as the overall analysis (S5 Fig in S1 File).

**Allergy.** In the meta-analysis for eczema, sixteen studies were included [53–68]. We found eczema cases had lower gut microbial alpha-diversity than controls (SMD:-0.34, 95%CI:-0.56, -0.12). All but two (Song et al and Reddel et al) of the studies included in the analysis were prospective birth cohorts in which children were followed until 6 months to 2.5 years of age and evaluated for a clinical diagnosis of eczema. We performed an analysis excluding

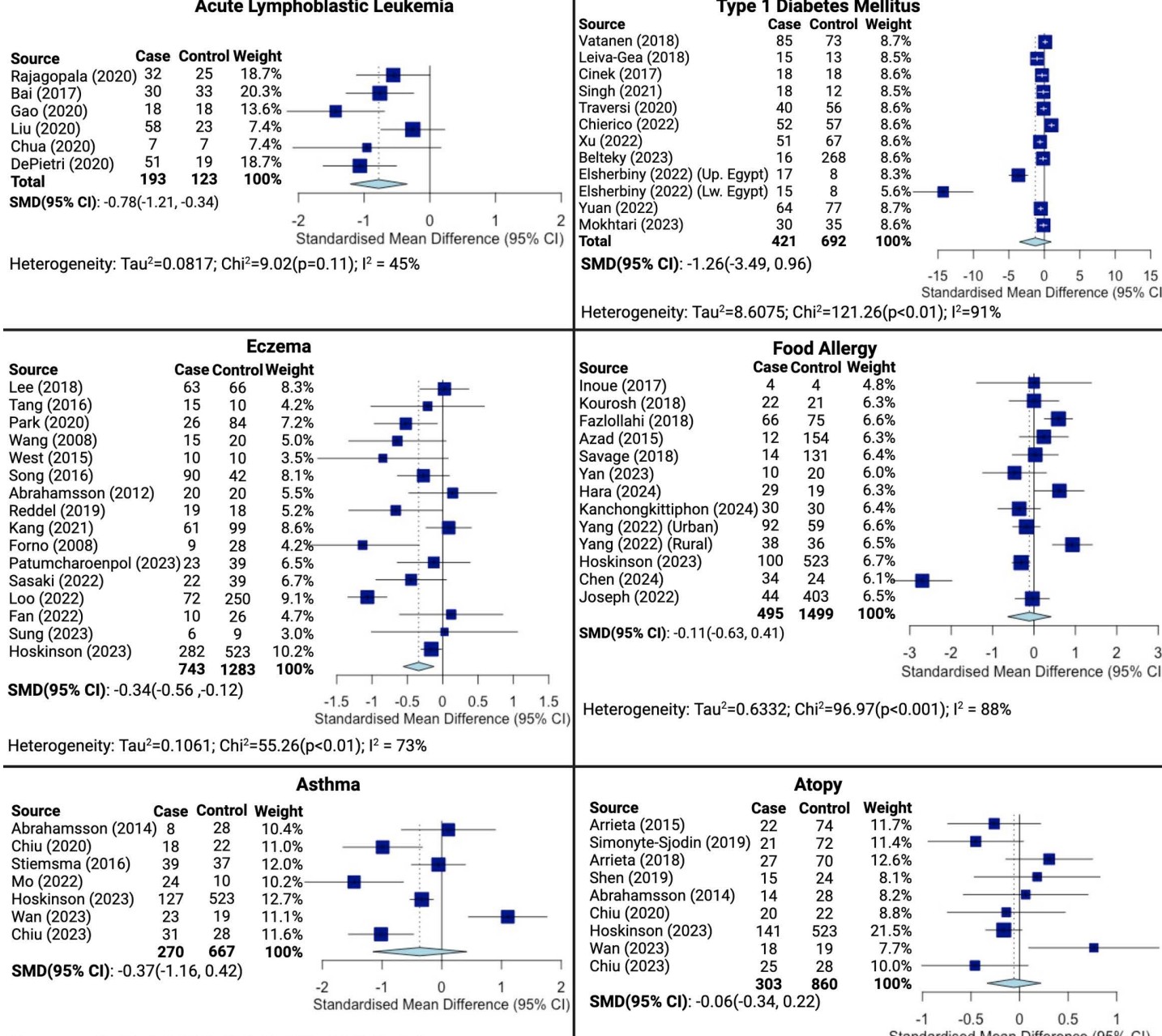

**Fig 2. Alpha-diversity (Shannon Index) in cases compared to controls: meta-analysis.** Abbreviations: SMD = standard mean difference, 95% CI = 95% confidence interval. Square data markers indicate the degree of difference in Shannon Index between cases and controls with the lines through the markers indicating 95% CIs. The size of the square data markers indicates the weight of the study. The diamond data marker indicates the overall pooled effect based on the included studies using a random effects model. Created with meta package in R and Biorender.com.

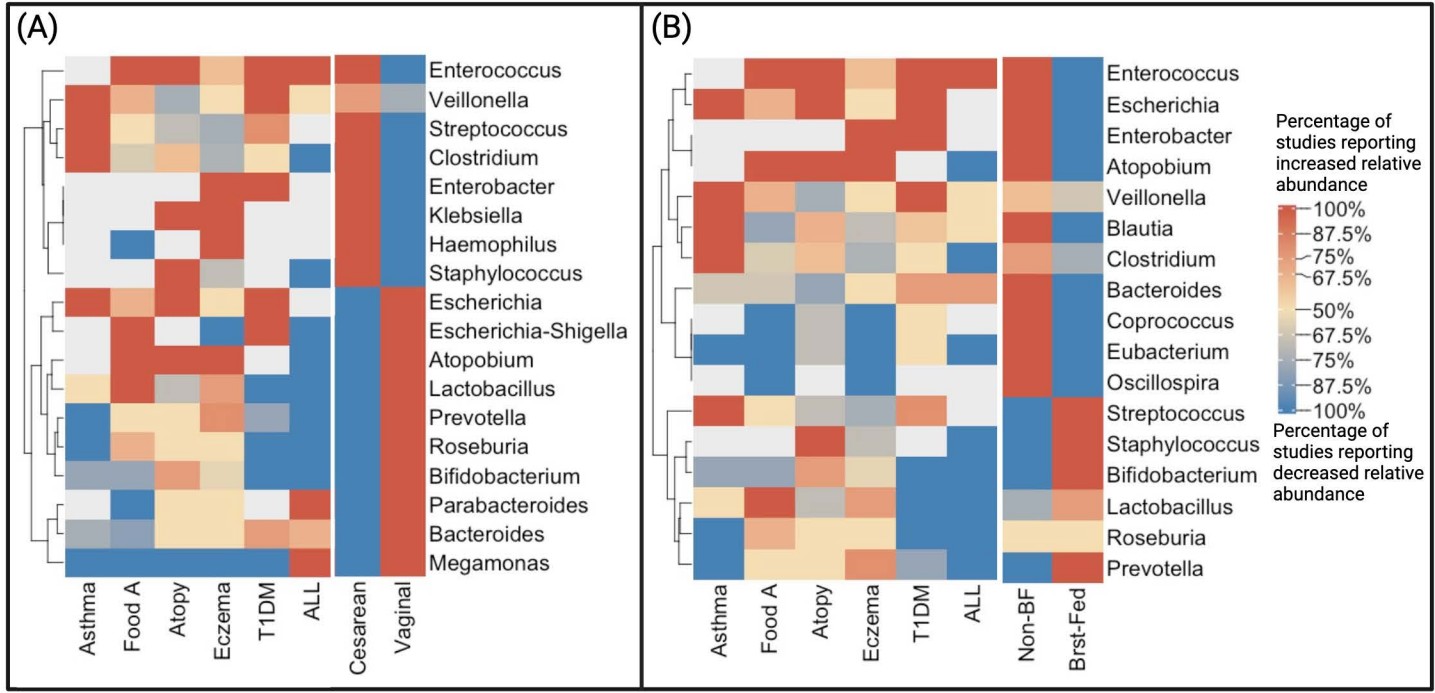

**Fig 3. Bacterial composition of disease types compared to described bacterial composition of delivery modes and breastfeeding status.**
Abbreviations: T1DM = type 1 diabetes mellitus, Cesarean = cesarean section delivery, Non-BF = non-breastfed, ALL = acute lymphoblastic leukemia, Food A = food allergy, Vaginal = vaginal delivery, Brst-Fed = breastfed. Clustering heatmap corresponding to the percentage of studies reporting increased or decreased relative abundance of cases compared to controls by disease types, delivery mode **(A)**, and breastfeeding status **(B)**. The horizontal direction are the disease types, delivery mode **(A)**, and breastfeeding status **(B)**, and the longitudinal direction are the bacterial genera. Studies reporting micro-biome composition by disease type (ALL, T1DM, eczema, atopy, asthma, and FA) were identified by systematic review. Studies reporting microbiome composition by delivery mode and breastfeeding status were identified by non-systematic literature review[89–96,98–102], and therefore are separated from the more highly curated disease-related studies on the left of each heatmap. Created with ComplexHeatmap package in R and Biorender.com.

those studies in which microbiome was not measured at or before diagnosis of eczema which yielded similar results to the overall analysis (S6 Fig in S1 File). Some of the infants in these studies received probiotics, but in those studies there was no difference in rates probiotic exposure between cases and controls [58,59,61]. An analysis excluding studies with probiotic exposure showed results were similar to the overall analysis (S7 Fig in S1 File). Though atopy [67,69–76] and asthma [67,73–77] tended to have lower alpha-diversity of the gut microbiome in cases compared to controls these results were not statistically significant (SMD:-0.06, 95%CI:-0.34, 0.22 and SMD:-0.37, 95%CI:-1.16, 0.42 respectively). In all of the included studies, diagnostic criteria for atopy included either a parental report of allergic symptoms or a clinical diagnosis and elevated antigen-specific IgE or positive skin prick testing. Subgroup analyses were performed to address timing of microbiome assessment, probiotic exposure, and inclusion based on parental history of allergy, and all sub-analyses showed similar results to the overall analysis. All but two studies (Wan et al, Chiu et al 2023) included in the atopy analysis were prospective cohorts following children from birth. Excluding these two studies revealed similar results to the overall analysis (S8 Fig in S1 File). In two studies, there was probiotic exposure in cases and controls with no significant difference in rates of exposure between groups [71,73]. Excluding these studies revealed no difference in alpha-diversity between cases and controls (S9 Fig in S1 File). Furthermore one study recruited cases and controls on the basis of a parental history of allergy [73]. Analysis excluding this study shows similar results to the overall analysis (S10 Fig in S1 File). Similar to atopy, alpha diversity of food allergy cases

was slightly lower than controls [67,78–88] though this was not statistically significant (SMD:-0.11, 95%CI:-0.63, 0.41). Food allergy was diagnosed with elevated food-specific IgE or positive skin prick testing for food antigens in all included studies. One study selected cases and controls based on a family history of allergy, though not specifically food allergy [82]. A second meta-analysis excluding this study was performed and yielded similar results to the overall meta-analysis (S11 Fig in S1 File). Only five studies measured microbiome composition at or before diagnosis of food allergy, and a subgroup analysis including only those studies shows similar results to the overall analysis with cases exhibiting lower alpha diversity than controls (S12 Fig in S1 File).

## Bacterial composition

Results from 77 studies were included in the qualitative analysis of bacterial composition of cases of various disease types compared to controls (S1 Table in S1 File). As delivery mode and breastfeeding are associated with childhood ALL, T1DM, and allergies and impact the gut microbiome [3,4,15,17–21,23,25,26], we compared the relative abundance of bacterial genera in cases versus controls to the bacterial composition associated with delivery mode and breastfeeding status in the literature [89–102].

Using a clustering analysis, we found that microbiome composition of ALL and T1DM are similar to that of C-section delivery and non-breastfed infants (Fig 3). Decreased abundance of lactobacillus is reported in ALL, T1DM, and atopy cases compared to controls as well as in CS versus vaginal delivery (VD), and non-breastfed vs breastfed infants (Fig 3). The relative abundance of Prevotella and Bifidobacterium is generally lower in cases of ALL, T1DM, and asthma compared to controls which is similar to CS and non-breastfed infants versus VD and breast-fed infants respectively (Fig 3). Furthermore, ALL, T1DM, eczema, atopy, food allergy, CS, and non-breastfed infants have higher relative abundance of Enterococcus compared to controls, VD, and breastfed infants (Fig 3). Eczema, T1DM, and CS have higher abundance of Enterobacter than their counterparts as well (Fig 3). Streptococcus and Staphylococcus are less abundant in eczema cases and non-breastfed infants (Fig 3). ALL, T1DM, and non-breastfed infants have in common a higher abundance of Bacteroides (Fig 3).

## Discussion

The results of our meta-analysis show a trend toward lower alpha-diversity of the gut microbiome in cases compared to controls among the three different disease types. In ALL and eczema the difference in alpha-diversity between cases and controls was statistically significant (Fig 2). In T1DM we found a similar trend, but this did not reach statistical significance (Fig 2). Several limitations affected the analysis of alpha diversity in T1DM cases compared to controls. One consideration is that two T1DM studies showing statistically significantly lower alpha-diversity in cases versus controls were excluded as appropriate summary statistics were not available [103,104]. Furthermore, two of the studies included in the meta-analysis (Vatanen et al, Cinek et al) were prospective studies in which patients were recruited based on having HLA haplotypes conferring a risk of T1DM development; therefore, the control population also carries a genetic risk of T1DM development which is not an ideal control group. An alternate meta-analysis was performed excluding studies whose control populations are genetically at risk for T1DM which resulted in a greater standard mean difference between cases and controls, though still not statistically significant (S2 Fig in S1 File). Finally, four studies included in the T1DM meta-analysis (Singh et al, Leiva-Gea et al, Elsherbiny et al, and Mokhtari et al) included subjects who were started on insulin therapy prior to measurement of microbiome which does not represent the ideal study design as insulin therapy and appropriate glycemic control could have an effect on the gut microbial composition. Finally, the timepoint of microbiome assessment varied across studies with some prospective studies assessing microbiome composition in infancy while case-control studies assessed composition at or near the time of diagnosis. It is possible that the microbiome alterations seen at diagnosis are not reflected in the microbiome in early infancy which may in part explain why in Vatanen et al cases exhibited higher diversity than controls.

Some of the studies in the eczema and atopy meta-analyses included probiotic exposure in both the cases and controls, and probiotic exposure is expected to alter the gut microbiome. However, there is no difference in rates of probiotic exposure in cases compared to controls in those individual studies, therefore it may not introduce bias into the meta-analysis results and sub-analyses excluding probiotic exposure are similar to the overall results.

Both our qualitative analysis of gut microbiome composition across disease types and meta-analysis are limited by the heterogeneity of the studies included in the analysis with regard to their design, methods, and timing of microbiome measurement.

The timepoint at which microbiome diversity and composition are measured is an important consideration when attempting to determine a potentially causal relationship between microbiome composition and disease. Longstanding disease or disease-directed treatment may itself alter microbiome composition clouding the picture. We have attempted to mitigate this limitation by performing subgroup analyses for each disease outcome excluding studies in which microbiome measurement did not occur at or before diagnosis when applicable (S4, S6, S8, S12 Figs in S1 File) with generally similar results to the overall analysis. Even so, this emphasizes the need for more prospective studies that measure microbiome at various timepoints from infancy through disease development to have a better understanding of how alterations in microbiome composition impact clinical outcomes.

There is also a great deal of variability in age at time of microbiome evaluation. As gut microbiome composition is dynamic in the first few years of life, age-associated microbiome compositional differences likely impact the results of this analysis thereby limiting the strength of the conclusions that can be drawn. Furthermore, the study design, techniques to quantify bacterial genera, and statistical methods differ among studies which also contributes the heterogeneity of the data in these analyses. Though our analysis uses a random effects model to account for some variability between studies, there may be biases that are not fully accounted for due to study heterogeneity. For example, many of the studies included in the qualitative analysis of microbiome composition report only statistically significant differences in bacterial abundance which may further introduce bias into our analysis. The diverse study designs and methodology of the available data also highlights the need for more uniformity in future studies evaluating the microbiome.

Our meta-analysis only includes alpha diversity which is just one of many important diversity measures. To gain a more complete understanding of the role of microbiome on the development of disease other measures of microbiome composition and diversity such as beta diversity should be evaluated as well. Beta diversity reflects the similarity of microbiome taxonomy which would be valuable in further exploring the association of microbiome diversity and these disease outcomes. In our review we found beta diversity to be reported less frequently than alpha diversity. Though other diversity measures will undoubtedly provide valuable information, a simplified measure such as alpha diversity could be quite useful in the application of a practical clinical screening tool in the future.

Diet and environmental exposures also have a significant impact on gut microbiome composition. As such, microbiome composition and diversity may differ by culture, ethnicity, socioeconomic status (SES), and geographic location as result of differing exposures and diets. Due to inconsistency in reporting ethnicity and SES, we were not able to adjust for these factors in our meta-analysis. However, this would be an important addition to future analyses. Though we could not formally assess ethnic and geographic factors in our analysis, it does include studies from many different countries and shows similar trends in microbiome diversity between cases and controls across different geographic locations.

Despite these limitations, the bacterial composition data we have analyzed is compatible with the suggestion from other recent studies on childhood ALL [105] and allergies [67]; the deficit in the microbiome of children with these conditions reflects an age associated functional immaturity. In the context of limited immune modulation in early life or priming by the microbiome [102,106] this could be crucial in disease risk escalation. We identified a trend toward lower microbial alpha-diversity in cases compared to controls in ALL, T1DM, eczema, atopy, asthma, and food allergy. It is intriguing that daycare attendance, breastfeeding, and vaginal delivery (compared to c-section) have been reported to have a protective

effect against the development of these childhood illnesses [18–21,107–112] and have been shown to affect the microbiome composition [89–101], suggesting a possible link between these exposures and disease. This observation is strengthened by the results of our clustering analysis showing that the microbial composition of patients with ALL, T1DM, and allergies appear more similar to that of infants delivered by c-section than vaginal delivery and non-breastfed infants versus those who are breastfed (Fig 3). Together the results of our analyses suggest that alterations in the microbiome may explain or mediate at least part of the association between these environmental factors and childhood ALL, T1DM, and allergies, highlighting the importance of early antigenic exposures to help prime the immune system leading to competent, adaptive immune responses later in childhood.

Though the association between the microbiome and disease shows promise as an important factor in development of ALL, T1DM, and allergies, other factors, such as genetics, are likely contributors which may be modified by microbiome composition [113]. Genetics and the environmentally-populated microbiome may work in concert in disease pathogenesis. Gene mutations along with common polymorphic variants and haplotypes are known to predispose to development of these diseases [15,24,46,114–118]. In the setting of harboring a distinctive, inherited susceptibility for ALL, T1DM, or allergies, alterations in the microbiome may be a shared risk factor to the development of overt disease. For instance, an ALL risk allele on *IKZF3* (rs2290400(T)) appears to be protective against T1DM and asthma, whereas the risk allele for T1DM and asthma in the same gene is associated with a decreased risk of childhood ALL [115]. Likewise, *IKZF1* has been identified as a causal gene in both T1DM and ALL, however the *IKZF1* ALL risk allele (rs10272724(C)) is protective for T1DM while conferring an increased risk of childhood ALL [116]. As these diseases are all linked to immune activation, this suggests that there are likely complex interactions leading to the development of disease including inverse genetic pleiotropy and potentially microbiome composition. Alteration of the microbiome leading to immune dysregulation represents a common risk factor for overt disease, whereas the underlying genetic susceptibilities of the individual dictate whether ALL, T1DM, or allergy develops.

The results of our study highlight the commonalities in the microbiome diversity and composition of children with ALL, T1DM, and allergies and implicate related mechanisms of disease development at least in the context of microbial diversity. The similarities between the gut microbial composition of cases, non-breastfed infants, and c-section delivered infants suggests that microbiome may be a link between these epidemiologic risk factors and ALL, T1DM, and allergies. Given that microbiome composition can be altered with early life exposures such as vaginal delivery, breastfeeding, and daycare exposure and interventions such as probiotics, diet, exercise, and fecal transplant, the gut microbiome is an accessible and modifiable risk factor for the prevention of disease. Systematic modification of the gut microbiome in early life faces significant challenges including selection of appropriate target populations, identification and standardized production of keynote bacterial species as safe probiotics and aspects of effective delivery. In early clinical studies, probiotics, dietary changes, and fecal transplants have been found to confer benefit for patients with T1DM and allergic disorders [119–123]. Several studies have shown improvement in eczema severity scores with probiotic administration [119,122,123]. Some have even shown decreased incidence of eczema when probiotics were administered in infancy as a preventative measure [122]. Similarly improvement in asthma severity as well as prevention have been reported with probiotic administration [122]. He et al reports two pediatric patients with T1DM who successfully achieved insulin independence after modulating their gut microbiome with fecal transplant [124]. These studies exhibit the practical application of gut microbiome alteration to modify disease. Furthermore, gut microbiome composition is also emerging as an early diagnostic tool as well as a means of identifying individuals predisposed to diseases such as type 2 diabetes mellitus and colorectal cancer [113,125]. If further investigations validate these findings, there could be a role for measuring microbiome diversity from stool samples as a screening tool to identify children at risk for ALL, T1DM, and allergies, and interventions to correct gut dysbiosis could be used for prevention or disease modification. Overall, these data encourage the notion of early life preventative strategies for childhood ALL, T1DM, and allergies.

## Supporting information

**S1 File. Supplementary tables, figures, and appendices including S1–S2 tables, S1–S12 figures, and S1–S3 appendices.**
(PDF)

**S2 File. Table containing all studies included and excluded from systematic review and meta-analysis.**
(XLSX)

**S3 Data. Table containing data extracted from studies included in qualitative and quantitative analyses.**
(XLSX)

**S4 Dataset. Statistical data required for meta-analyses as well as qualitative analyses.**
(XLSX)

## Author contributions

**Conceptualization:** Mel Greaves.

**Formal analysis:** Rachel Gallant.

**Investigation:** Rachel Gallant, Samiha Reza.

**Methodology:** Rachel Gallant, Samiha Reza, Joseph L. Wiemels, Mel Greaves.

**Supervision:** Joseph L. Wiemels, Mel Greaves.

**Visualization:** Rachel Gallant.

**Writing – original draft:** Rachel Gallant.

**Writing – review & editing:** Samiha Reza, Joseph L. Wiemels, Mel Greaves.

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
