## [Decision Letter · Decision Letter 0]

26 Jan 2025

PONE-D-24-51052Microbiome and pediatric leukemia, diabetes, and allergies: systematic review and meta-analysisPLOS ONE

Dear Dr. Gallant,

Thank you for submitting your manuscript to PLOS ONE. After careful consideration, we feel that it has merit but does not fully meet PLOS ONE’s publication criteria as it currently stands. Therefore, we invite you to submit a revised version of the manuscript that addresses the points raised during the review process.

Please revise the manuscript as per reviewers suggestion, ultimately to reviewer 1. 

We look forward to receiving your revised manuscript.

Kind regards,

Fahrul Nurkolis

Academic Editor

PLOS ONE

“This work was supported by Cancer Research UK (CRM 171X) (MG).”

3. We note that your Data Availability Statement is currently as follows: [All data is derived from previously published work and is publicly available.]

4. As required by our policy on Data Availability, please ensure your manuscript or supplementary information includes the following:

Reviewers' comments:

Reviewer's Responses to Questions

**Comments to the Author**

1. Is the manuscript technically sound, and do the data support the conclusions?

Reviewer #1: Partly

Reviewer #2: Yes

2. Has the statistical analysis been performed appropriately and rigorously? 

Reviewer #1: No

Reviewer #2: Yes

3. Have the authors made all data underlying the findings in their manuscript fully available?

Reviewer #1: Yes

Reviewer #2: Yes

4. Is the manuscript presented in an intelligible fashion and written in standard English?

Reviewer #1: Yes

Reviewer #2: Yes

5. Review Comments to the Author

Reviewer #1: After reviewing the manuscript titled "Microbiome and pediatric leukemia, diabetes, and allergies: systematic review and meta-analysis", here are several weaknesses and areas for improvement:

The manuscript effectively addresses an important research area but could be improved in several aspects:

1. While the manuscript includes a robust meta-analysis, the included studies exhibit substantial heterogeneity in design, microbiome measurement methods, and timing of microbiome assessment. This variability weakens the strength of the conclusions, as the lack of uniformity may introduce biases or confounders that are not adequately accounted for.

2. Limiting the included studies to English-only publications may introduce language bias, potentially excluding relevant high-quality research published in other languages.

3. Although the study discusses confounders such as antibiotic use, breastfeeding, and delivery mode, these factors are not uniformly addressed across the included studies. A more rigorous subgroup analysis or adjustment for these confounders would enhance the reliability of the findings.

4. The meta-analysis focuses exclusively on alpha diversity metrics (e.g., Shannon Index). While this is an important measure, other diversity indices and taxonomic composition metrics, such as beta diversity, could provide deeper insights into microbiome changes.

5. Many studies included are cross-sectional. A greater emphasis on longitudinal studies would provide a better understanding of causality and temporal dynamics of the microbiome in disease progression.

6. Although funnel plots were used to assess publication bias, these may not fully capture biases inherent to microbiome research, such as selective reporting of positive findings. This should be discussed further.

7. There is variability in the demographic details reported across studies, including ethnicity, socioeconomic factors, and geographic location. These factors significantly impact microbiome composition and should be more systematically accounted for.

8. Some results, particularly for T1DM, fail to reach statistical significance, yet are discussed as trends. This could mislead readers into overestimating the strength of these associations.

9. The clustering analysis comparing disease-related microbiome profiles with delivery mode and breastfeeding status is intriguing but appears speculative without robust statistical validation. Including additional data to support these claims would improve the scientific rigor.

10. The manuscript highlights the potential for microbiome-based interventions but fails to provide detailed, evidence-based recommendations or an actionable framework for implementing such strategies in clinical practice.

Addressing these issues would enhance the manuscript's clarity, scientific rigor, and impact. Future iterations should strive for a more standardized approach to data inclusion and analysis, and further explore longitudinal, diverse, and multi-factorial datasets to strengthen the findings.

Reviewer #2: This manuscript offers an insightful and timely exploration of the relationship between gut microbiome diversity and pediatric conditions like acute lymphoblastic leukemia (ALL), type 1 diabetes mellitus (T1DM), and allergic disorders. It’s clear that a lot of thought and effort went into this work, and it adheres to rigorous scientific standards. The authors followed PRISMA guidelines carefully, and the inclusion and exclusion criteria are well-justified. The findings—particularly the link between reduced microbiome diversity and conditions like ALL and eczema—are compelling and well-supported by the data. I also appreciate the use of subgroup analyses and sensitivity tests to address key confounders, like antibiotic use or insulin therapy, which strengthens the validity of the conclusions. That said, while trends were observed for T1DM and asthma, the lack of statistical significance here deserves a little more reflection on the possible limitations.

The statistical approach was thoughtful and thorough. The random-effects model used for the meta-analysis was appropriate given the variability across studies, and funnel plots helped confirm that publication bias wasn’t an issue. Using the Shannon Index as a measure of alpha diversity was a smart choice, as it’s both widely recognized and relevant. I did wonder, though, whether incorporating meta-regression might have added even more depth—especially for teasing apart how factors like study design or age at diagnosis could influence the results. Additionally, the wide confidence intervals in some cases, such as for T1DM, suggest there might be more to discuss regarding the power or variability of certain analyses.

In terms of writing, the manuscript is clear and easy to follow, though a few areas might benefit from some streamlining. The Methods and Results sections, in particular, are pretty dense with statistical details, which might make them tough for readers who aren’t deeply familiar with meta-analysis. Simplifying some of this language or adding a brief summary in plain terms could make these sections more approachable. The Discussion is well thought out and comprehensive, but some of the transitions—especially between microbiome findings and the interplay with genetic or environmental factors—could flow a bit more smoothly. A little more context around some of the more technical methods, like clustering analysis or the use of the ComplexHeatmap package, would also help make this work more accessible to a wider audience.

The authors have done an excellent job ensuring data transparency. They clearly state that all data used are from publicly available sources, with summary statistics compiled and available upon request. Even for studies where data had to be estimated from graphical representations, the process is well-documented, which is reassuring. This level of transparency is commendable and aligns with best practices in systematic reviews.

Overall, this study makes a strong contribution to the growing body of research on the microbiome’s role in childhood diseases. The connections drawn between early-life exposures, microbiome diversity, and disease risk are fascinating and highlight exciting possibilities for future research. The idea of using microbiome-based interventions, like probiotics or dietary modifications, as preventative strategies is particularly intriguing. I’d encourage the authors to dive a little deeper into the practical challenges and limitations of applying these findings clinically. For instance, how feasible is it to use microbiome diversity as a screening tool, or how realistic are these interventions in a real-world healthcare setting?

In summary, this is an impressive piece of work that brings meaningful insights to an important area of research. A few tweaks to improve accessibility, clarify some transitions, and expand on the clinical implications would make it even stronger. I look forward to seeing how this research develops and its potential impact on pediatric care.

6. PLOS authors have the option to publish the peer review history of their article (what does this mean? ). If published, this will include your full peer review and any attached files.

**Do you want your identity to be public for this peer review?** For information about this choice, including consent withdrawal, please see our Privacy Policy .

Reviewer #1: No

Reviewer #2: No

---

## [Author Response · Author response to Decision Letter 0]

5 Mar 2025

Dear Editors and Reviewers,

We thank you for your feedback on our manuscript “Microbiome and pediatric leukemia, diabetes, and allergies: systematic review and meta-analysis”. We acknowledge the valuable points made by the editors and reviewers and in response have made changes that improve the manuscript. The field of microbiome does have some limitations in terms of study design and organisms studied and reported. These limitations, in turn, introduce challenges to a systematic review of the published literature. However, based on the reviewer comments we have further addressed these limitations with sub-analyses and expanded discussion that we feel has strengthened the analysis and manuscript overall.

We appreciate your thoughtful review of this systematic review and meta-analysis and look forward to your further consideration.

Sincerely,

Rachel Gallant MD, MS

Journal Requirements

Response: The title page and body of the manuscript have been updated to be compliant with PLOS ONE’s style requirements.

“This work was supported by Cancer Research UK (CRM 171X) (MG).”

Response: The funding statement has been included in the cover letter and confirms that the funders had no role in study design, data collection and analysis, decision to publish, or preparation of the manuscript.

3. We note that your Data Availability Statement is currently as follows: [All data is derived from previously published work and is publicly available.]

Response: The data extracted from the published studies included the quantitative and qualitative analyses has been compiled and the data set included in the Supporting Information. The data sharing statement has been updated at the end of the manuscript.

4. As required by our policy on Data Availability, please ensure your manuscript or supplementary information includes the following:

If applicable for your analysis, a table showing the completed risk of bias and quality/certainty assessments for each study or outcome. Please ensure this is provided for each domain or parameter assessed. For example, if you used the Cochrane risk-of-bias tool for randomized trials, provide answers to each of the signaling questions for each study. If you used GRADE to assess certainty of evidence, provide judgements about each of the quality of evidence factor. This should be provided for each outcome.

Response: Tables containing a list of included and excluded studies (with reason for exclusion) have been included in Supporting Information. All extracted data is included in a Supporting Information table including confirmation of eligibility, data extractor, extraction date, and source of data if other than publication. All data required to replicate quantitative and qualitative analyses are included in a Supporting Information table. GRADE assessments are included in Supporting Information.

Reviewer Comments

Reviewer #1: After reviewing the manuscript titled "Microbiome and pediatric leukemia, diabetes, and allergies: systematic review and meta-analysis", here are several weaknesses and areas for improvement:

The manuscript effectively addresses an important research area but could be improved in several aspects:

1. While the manuscript includes a robust meta-analysis, the included studies exhibit substantial heterogeneity in design, microbiome measurement methods, and timing of microbiome assessment. This variability weakens the strength of the conclusions, as the lack of uniformity may introduce biases or confounders that are not adequately accounted for.

Response: We agree with the reviewer that the heterogeneity in study design, methods, and timing of microbiome assessment may introduce bias or confounders that weaken the strength of conclusions drawn from this analysis. However, we addressed this by using a random effects model for the meta-analysis and performing subgroup analyses to account for differences in timing of microbiome measurement and study design as described in the Methods section and Supplemental Materials. Furthermore, we have improved the discussion of these limitations in the Discussion section of the manuscript (lines 362-367).

2. Limiting the included studies to English-only publications may introduce language bias, potentially excluding relevant high-quality research published in other languages.

Response: We agree that in principle, restricting to English-only studies could introduce language bias. However, the titles and abstracts were available in English for all of the papers reviewed. As such, even non-English manuscripts were screened as part of the title and abstract screen. There was only one paper that was excluded solely because it was not available in English, the other four non-English manuscripts that initially passed the title and abstract screen were further evaluated and were found to have the wrong study design and therefore were also not eligible for inclusion for that reason as well. Therefore, though our inclusion criteria was limited to English-only, for this systematic review there was not a significant proportion of studies excluded solely based on language (1/6439, 0.02% of studies excluded). The flow chart of study selection in Figure 1 has been updated to better reflect this.

3. Although the study discusses confounders such as antibiotic use, breastfeeding, and delivery mode, these factors are not uniformly addressed across the included studies. A more rigorous subgroup analysis or adjustment for these confounders would enhance the reliability of the findings.

Response: When developing the protocol for this systematic review, we intended to collect any available variables that could affect the gut microbiome such as antibiotic use, breastfeeding, and delivery mode. However, most of the studies identified in the literature search did not include data about exposures. Most of the included studies only excluded participants with recent antibiotic exposure. For those studies that did report recent antibiotic use, we performed subgroup analyses excluding those with antibiotic exposure as described in the manuscript in the Results section under “ALL”(lines 199-202) and in Supplemental Figure S2.

4. The meta-analysis focuses exclusively on alpha diversity metrics (e.g., Shannon Index). While this is an important measure, other diversity indices and taxonomic composition metrics, such as beta diversity, could provide deeper insights into microbiome changes.

Response: We agree that additional diversity measures would be a valuable addition to future analyses. Of the studies identified in this review, alpha diversity was by far reported most commonly. As more and more studies are reporting beta diversity, an opportunity to include beta diversity in a formal analysis would likely be feasible in the future. Beta diversity does help reveal compositional differences in the microbiomes of subgroups of samples which could be very relevant to the underlying biology, for example, with immune priming in infancy. However, for this to be informative at the bacterial species level would require shotgun sequencing or metagenomics. Most of the reported results included in our analysis are derived from 16S amplification which is reliable at the genus level but less so for species. Alpha diversity is still a valuable measure and given its simplicity could potentially be useful in the application of clinical screening tool in the future. We have expanded upon this in the Discussion section of the manuscript (lines 374-388).

5. Many studies included are cross-sectional. A greater emphasis on longitudinal studies would provide a better understanding of causality and temporal dynamics of the microbiome in disease progression.

Response: We agree that the cross-sectional study design is not the ideal study type for determining a causal relationship between microbiome composition and disease outcome as microbiome composition may be impacted by the disease process or its treatment. As such, cross sectional studies may include participants who have had long standing disease or those who have already initiated treatment both of which may not reflect how dysbiosis could cause these disorders. We have addressed this by performing subgroup analyses excluding studies where microbiome measurements were not taken prior to or at diagnosis. T1DM subgroup analysis is shown in Figure S4 in the Supporting Information. Additional analyses were performed for disease outcomes of eczema, atopy, and food allergy excluding studies in which microbiome was not measured prior to or at diagnosis which have been including in the Results section and Supporting Information (Supporting Information Figures S6, S8, S12). Further expansion is included in the Discussion as well (lines 348-361).

6. Although funnel plots were used to assess publication bias, these may not fully capture biases inherent to microbiome research, such as selective reporting of positive findings. This should be discussed further.

Response: We agree with the reviewer in that there are likely more sources of potential bias in microbiome research particularly in the reporting of bacterial abundance. By use of funnel plots and GRADE we hope to minimize bias in our quantitative analysis, however we acknowledge that microbiome research may have inherent biases that we cannot account for particularly in the qualitative analysis. We have added more discussion about the risk for bias and limitations into the discussion section (lines 367-371).

7. There is variability in the demographic details reported across studies, including ethnicity, socioeconomic factors, and geographic location. These factors significantly impact microbiome composition and should be more systematically accounted for.

Response: As diet and environmental exposures have a significant impact on the gut microbiome, ethnicity, socioeconomic factors, and geographic location will all impact microbiome composition. As ethnicity and socioeconomic status are not consistently reported in the studies included in this systematic review, it is difficult to account for these in the meta-analysis. We agree that this should be a focus in future studies. We have expanded upon this topic in the Discussion section of the manuscript (lines 389-396).

8. Some results, particularly for T1DM, fail to reach statistical significance, yet are discussed as trends. This could mislead readers into overestimating the strength of these associations.

Response: We have changed the wording to make it clear that there is not a statistically significant difference in microbiome diversity in cases compared to controls specifically for T1DM, atopy, and asthma (lines 213-214).

9. The clustering analysis comparing disease-related microbiome profiles with delivery mode and breastfeeding status is intriguing but appears speculative without robust statistical validation. Including additional data to support these claims would improve the scientific rigor.

Response: Due to the inconsistency of reporting between studies, we were not able to do a robust statistical analysis with this data. As such, we performed an exploratory qualitative analysis which we have displayed visually in a heatmap. We have included greater detail describing how this analysis was performed in the Methods section as well as added an additional reference for the ComplexHeatmap package in R to better describe the statistical methods (lines 168-172).

10. The manuscript highlights the potential for microbiome-based interventions but fails to provide detailed, evidence-based recommendations or an actionable framework for implementing such strategies in clinical practice.

Response: We have enhanced the discussion of practical applications of microbiome modification to augment disease severity and treatment as well as using microbiome as a screening tool for those at risk for disease. This has been added to the Discussion section (lines 443-466).

Addressing these issues would enhance the manuscript's clarity, scientific rigor, and impact. Future iterations should strive for a more standardized approach to data inclusion and analysis, and further explore longitudinal, diverse, and multi-factorial datasets to strengthen the findings.

Reviewer #2: This manuscript offers an insightful and timely exploration of the relationship between gut microbiome diversity and pediatric conditions like acute lympho

---

## [Editor Report · Decision Letter 1]

22 Apr 2025

Microbiome and pediatric leukemia, diabetes, and allergies: systematic review and meta-analysis

PONE-D-24-51052R1

Dear Dr. Gallant,

We’re pleased to inform you that your manuscript has been judged scientifically suitable for publication and will be formally accepted for publication once it meets all outstanding technical requirements.

Kind regards,

Fahrul Nurkolis

Academic Editor

PLOS ONE
---

## [Editor Report · Acceptance letter]

PONE-D-24-51052R1

PLOS ONE

Dear Dr. Gallant,

I'm pleased to inform you that your manuscript has been deemed suitable for publication in PLOS ONE. Congratulations! Your manuscript is now being handed over to our production team.

Kind regards,

on behalf of

Dr. Fahrul Nurkolis

Academic Editor

PLOS ONE